# The Cytokine CX3CL1 and ADAMs/MMPs in Concerted Cross-Talk Influencing Neurodegenerative Diseases

**DOI:** 10.3390/ijms24098026

**Published:** 2023-04-28

**Authors:** Matilda Iemmolo, Giulio Ghersi, Giulia Bivona

**Affiliations:** 1Department of Biological, Chemical and Pharmaceutical Sciences and Technologies (STEBICEF), University of Palermo, 90128 Palermo, Italy; matilda.iemmolo@unipa.it; 2Department of Biomedicine, Neurosciences and Advanced Diagnostics, Institute of Clinical Biochemistry, Clinical Molecular Medicine and Laboratory Medicine, University of Palermo, 90133 Palermo, Italy; giulia.bivona@unipa.it

**Keywords:** CX3CL1, Fraktaline, ADAM’s, neurodegeneration, MMP’s

## Abstract

Neuroinflammation plays a fundamental role in the development and progression of neurodegenerative diseases. It could therefore be said that neuroinflammation in neurodegenerative pathologies is not a consequence but a cause of them and could represent a therapeutic target of neuronal degeneration. CX3CL1 and several proteases (ADAMs/MMPs) are strongly involved in the inflammatory pathways of these neurodegenerative pathologies with multiple effects. On the one hand, ADAMs have neuroprotective and anti-apoptotic effects; on the other hand, they target cytokines and chemokines, thus causing inflammatory processes and, consequently, neurodegeneration. CX3CL1 itself is a cytokine substrate for the ADAM, ADAM17, which cleaves and releases it in a soluble isoform (sCX3CL1). CX3CL1, as an adhesion molecule, on the one hand, plays an inhibiting role in the pro-inflammatory response in the central nervous system (CNS) and shows neuroprotective effects by binding its membrane receptor (CX3CR1) present into microglia cells and maintaining them in a quiescent state; on the other hand, the sCX3CL1 isoform seems to promote neurodegeneration. In this review, the dual roles of CX3CL1 and ADAMs/MMPs in different neurodegenerative diseases, such as Alzheimer’s disease (AD), Parkinson’s disease (PD), Huntington’s disease (MH), and multiple sclerosis (MS), are investigated.

## 1. Introduction

Chemokines are a large group of low molecular weight proteins of the cytokine family. Chemokines are small chemotactic proteins that call inflammatory cells to the site of inflammation. For this reason, chemokines and their receptors represent therapeutic targets in inflammatory disorders. In humans, there are more than fifty closed chemokines, small heparin-binding proteins (8–10 kDa) identified by their chemotactic activity in bone marrow-derived cells. There are two chemokines: inflammatory chemokines, which recruit leukocytes in response to physiological stress, and homeostatic chemokines, which explain the trafficking of basal leukocytes and the formation of the architecture of secondary lymphoid organs. The inflammatory activity of chemokines can be caused by stimuli that alter cellular homeostasis, such as infections, immune disorders, and pathologies [1,2]. Chemokine receptors are bound to the cell membrane through G-protein-coupled seven-transmembrane helical segments that transduce intracellular signaling [1,2].

The cytokines are subdivided into four subgroups C, CC, CXC, and CX3C [3,4]. These subgroups are characterized by the number of interspersed amino acids between the first two cysteine residues in the primary structure. Thus, the CX3C subgroup contains three interspersed, non-conserved amino acids. Like the other three subgroups, the CX3C subgroup contains conserved cysteine residues at positions 8, 12, 34, and 50 in humans and other species. The CX3C subgroup consists of only one known member, CX3CL1 [5]. CX3CL1, also named Fraktaline (FKN), exists in two forms: as a transmembrane protein that acts as an adhesion molecule, interacting with its unique CX3CR1 receptor, and as a soluble form that is generated by a proteolytic process that acts as a signaling molecule [6,7]. The form of a transmembrane protein is composed of 373 total amino acids consisting of an extracellular N-terminal domain (residues 1–76), a mucin-like stalk (residues 77–317), a transmembrane α helix (residues 318–336), and a short cytoplasmic tail (residues 337–373) [8]. The soluble form is generated by different proteolytic processes like a proteolytic cleavage by A Disintegrin and metalloproteinase domain-containing protein (ADAM)-10 that generates various soluble forms of CX3CL1 [9,10], instead of inflammatory agents, such as lipopolysaccharide (LPS), interleukin (IL)-1β, or phorbol esters, enhance the cleavage of CX3CL1 through tumor necrosis factor (TNFα-converting enzyme, ADAM17) [11] (Figure 1). Therefore, also metalloproteases, thanks to their proteolytic activity, play a fundamental role in the neuroinflammatory process in which CX3CL1 is involved. The proteolytic release of membrane-bound proteins, such as fraktaline, generally allows cells to act, communicate and respond to their microenvironment. ADAM endopeptidases (a disintegrin and metalloproteinase) are an abundantly diverse family of type I transmembrane metalloproteinases that hydrolyze adhesion molecules, signaling receptors, and growth factors. The human genome encodes 22 different ADAMs, essential for numerous developmental events, including CNS development [12,13]. Moreover, the ADAMs are regulated by TIMPs that inhibit and regulate the release of cell-membrane proteins, a process known as ectodomain shedding [14]. Furthermore, dysregulation of ADAM activity is implicated in various pathophysiological states, such as asthma, infertility, Alzheimer’s disease, and cancer. ADAMs are single-step transmembrane proteins with a modular domain organization. ADAM10 and ADAM17 have an ectodomain containing an N-terminal signal sequence and an adjacent pro-domain, followed by domains rich in metalloproteinases, disintegrin, and cysteine. The full-length ADAM precursors are catalytically inactive when the prodomain is intact and associated with the mature enzyme. Their cysteine and disintegrin-rich domains also play a self-regulating role by suppressing metalloproteinase activity in the absence of substrates. These domains can also direct enzymatic activity by directly binding proteolytic substrates [15]. Recent findings also show that CX3CL1 is a substrate for beta-secretase (BACE1) and gamma-secretase, which are also involved in the production of amyloid β (Aβ) from amyloid precursor protein (APP) cleavage [16]. This proteolytic processing of CX3CL1 appears to regulate its function.

## 2. A Different Form of CX3CL1

As previously mentioned, fraktaline exists in two forms: a transmembrane form bonded to the membrane, which acts as an adhesion molecule, and a soluble form that could take on different functions. The Fraktaline is synthesized as a precursor of 50–75 kDa, which undergoes rapid maturation, produced by glycosylation, to yield per mature FKN of 100 kDa. These data suggest that FKN is initially synthesized as an intracellular precursor that undergoes glycosylation and transport to the cell surface as a 100 kDa glycoprotein. The 100 kDa FKN can then be released from the cell surface, producing a soluble 85 kDa fragment that probably contains the majority of the glycosylated ectodomain and a transmembrane cytoplasmic tail fragment of ≥20 kDa [17]. It is known that the activity of different proteases mediates the cleavage of CX3CL1. There is evidence that metalloproteinases of the ADAM family also play a role in the shedding of CX3CL1. The cleavage of CX3CL1 is mediated by different members of the ADAM family depending on whether the cleavage is constitutive or induced. It has been shown through inhibition studies with natural inhibitors of TIMP-1 and TIMP-2 metalloproteinases that the ADAM10 protease is a candidate for constitutive cleavage of CX3CL1. TIMP-1 inhibited constitutive cleavage, while TIMP-2 did not affect constitutive cleavage (this excludes the involvement of matrix-type MMPs (MT MMP types 1, 2, and 3) in the shedding of CX3CL1, as these proteases are blocked from TIMP-2). This thesis was also confirmed by the overexpression of this protease, which led to an increase in the constitutive cleavage of CX3CL1 and also by the deletion of the ADAM10 gene, which abolished an important part of the constitutive shedding of CX3CL1 [9]. In contrast, the induced shedding of CX3CL1 appears to be mediated by another protease of the ADAM family, namely ADAM17/TACE. Induction with PMA (phorbol-12-myristate-13-acetate) stimulated the cleavage of CX3CL1 by TACE but not by ADAM10; this was confirmed by the fact that the inhibition of TACE but not the inhibition of ADAM10 blocked the spread of CX3CL1 in several types of PMA-stimulated cell lines. Taken together, both ADAM10 and TACE cut CX3CL1, but ADAM10 contributes solely to constitutive cleavage, while TACE exclusively mediates inducible shedding [17]. Furthermore, the expression of TACE is upregulated in response to inflammatory stimuli in vivo, suggesting that TACE is activated during inflammatory responses where it can mediate the dispersion of a diverse set of proteins, including the circulation of TNF-α and fraktaline itself [18,19]. Cell adhesion tests were performed to understand the consequences of cleavage of the CX3CL1 protease, after which the importance of the CX3CL1 membrane for maintaining cell adhesion was demonstrated. The mixed inhibitor of TACE/ADAM10 has been shown to block deadhesion with the same potency as the selective inhibitor of ADAM10, suggesting that ADAM10 rather than TACE is relevant for promoting cell deadhesion [9].

CX3CL1 is also a potential substrate of β-secretase and γ-secretase, following cleavage by α-secretase. The overexpression of BACE1 generated two C-terminal fragments at low m.w. (12 kDa and 10 kDa) and three fragments at higher m.w., like the sequential cleavage of the amyloid precursor protein (APP) from α- and β -secretase (BACE1). Confirmation that these two C-terminal fragments were generated by BACE1 activity was established by treatment with a potent BACE inhibitor (BACE1 IV inhibitor) which decreased the levels of these two fragments [16]. Similarly, it has been shown that the intracellular domain of CX3CL1 undergoes cleavage by γ-secretase, using two different inhibitors for the γ-secretase complex and PSEN 1 and 2 deficient embryonic fibroblasts. γ-secretase occurs within the cell membrane and leads to the generation of smaller fragments released from the cell membrane; these fragments became detectable following the inhibition of proteasome degradation but were absent when the secretase activity was blocked. Thus, the cleavage of CX3CL1 by γ-secretase can be rapidly followed by proteasomal degradation of the resulting fragments in the cytoplasm. This could play a mechanism to free the cell membrane from C-terminal fragments of transmembrane chemokines that would otherwise accumulate in the cell membrane following the continuous detachment of life-size molecules [20].

At the same time, however, as shown for Notch and also E-cadherin, the cleavage of CX3CL1 by the γ-secretase complex could eventually lead to the generation of intracellular fragments implicated in intracellular signaling [21]. Fan et al. have shown that the intracellular domain of CX3CL1, following the cleavage from the γ-secretase complex, undergoes a nuclear translocation by altering gene expression. Indeed, it was revealed that CX3CL1 controls TGFβ/Smad signaling pathways through its C-terminal intracellular domain, which cannot bind CX3CR1 due to a lack of epidermal growth factor-like domain and logical constraints. The C-terminal intracellular domain of CX3CL1 induces the expression of many genes directly or indirectly through the effects of the TGFβ superfamily. One such molecule is Pax6, which is known to be expressed by radial glial progenitors and is required for the unidirectionality of lineage commitment to neuronal differentiation [22]. Furthermore, this domain controls the Smad pathway, and Smad2-mediated transcription is critical for brain development, function, and/or maintenance of neuronal cells [23]; phospho-Smad2 appears to be an effector of this path [24]. Thus, it can be said that the C-terminal intracellular domain of CX3CL1 generated by the cleavage of the γ-secretase complex promotes neurogenesis.

The contribution of cell membrane–expressed CX3CL1 or its soluble variant that is generated by proteolytic shedding of the transmembrane molecule is not yet clear. Aimee N. Winter et al. evaluated the function of membrane CX3CL1 (mCX3CL1) and the soluble form (sCX3CL1) in CX3CL1 -/- mice. The knockout mice showed deficits in long-term memory, spatial learning, and motor performance. These related alterations were deficient in both hippocampal neurogenesis and Long Term Potentiation (LTP). Treatment of CX3CL1 -/- mice with sCX3CL1 improved memory and spatial learning deficits, suggesting that sCX3CL1 activity is particularly important for performing hippocampus-dependent associative learning and memory tasks. mCX3CL1 also improved spatial learning and memory in a similar way to sCX3CL1; this could indicate that the transmembrane shape has a specific role in the formation of spatial memory, such as the ability to improve function within the dentate gyrus. As far as neurogenesis is concerned, it is interesting to note that only sCX3CL1 seems to save the neurogenesis of the hippocampus. The treatment restored the expression of both Ki67 and DCX in the subgranular zone (SGZ), indicative of greater neurogenesis. Likewise, only the soluble form of CX3CL1 restored LTP. Regarding motor performance, sCX3CL1 has been shown to restore the neural pathways associated with motor function, while mCX3CL1 does not seem to verify this process [25]. Furthermore, Gunner et al. demonstrated that sCX3CL1 signaling is required for activity-dependent synaptic remodeling that occurs in the cortex following sensory injury. This supports the idea that membrane-bound forms of CX3CL1 may not play a significant role in the remodeling of synaptic circuits, a process that may be involved here in normalizing motor function, and suggests instead that this synaptic remodeling may be predominantly mediated by sCX3CL1 signaling [26].

## 3. CX3CL1 and ADAMs/MMPs Involved in Neuroinflammation

CX3CL1 is primarily expressed by epithelial cells in the lung, kidney, and intestine and is creased in endothelial cells at sites of local inflammation, but CX3CL1 expression is higher in the brain than in most other organs [27]. In the central nervous system (CNS), CX3CL1 is widely expressed by neurons within the hippocampus and cortex and interacts with its unique CX3CR1 receptor on the microglia. Therefore, the interaction between neurons and microglia is mediated by CX3CL1/CX3CR1 signaling. CX3CL1/CX3CR1 signaling keeps microglia in a resting state (quiescent or inactivated), thereby inhibiting the release of pro-inflammatory cytokines. Furthermore, it enhances the phagocytosis of pre-synaptic elements [28]. The molecular mechanisms underlying microglial responses to CX3CL1 involve transient calcium mobilization, chemotaxis, matrix metalloproteinase production 2 and 9, and a cascade of protein phosphorylation and enzymatic activation [29]. Several studies have shown that the binding of fraktaline to CX3CR1 induces MAP kinase activation. There are three cascades of MAP kinases: c-Jun N-terminal kinase/stress-activated protein kinase (JNK/SAPK), p38 and extracellular signal-bound kinase (ERK1/ERK2). JNK/SAPK signaling is induced by exposure to ultraviolet radiation, heat shock, or inflammatory cytokines. The p38 pathway is activated in response to inflammatory cytokines, endotoxins, and osmotic stress. The ERK pathway is stimulated upon binding extracellular growth factors to tyrosine kinase-bound receptors. Furthermore, in microglia, CX3CR1 is involved in intracellular signaling pathways, such as phospholipase C (PLC), PI3K, and ERK, by recruiting transcription factors, such as NF-κB and cyclic adenosine monophosphate response element-binding protein (CREB) [30]. The CX3CL1/CX3CR1 pathway plays a pivotal role in neuroinflammation (Figure 2). Cardona et al. have shown that the increase in the activity of microglia due to an interruption of the signaling of fraktaline with its receptor is accompanied by an increase in neuronal death. On the contrary, exposure of microglia to CX3CL1 reduces the toxicity of microglia and protects neurons from apoptosis in inflammatory conditions. Furthermore, because microglia react quickly to protect normal neurons and eliminate damaged neurons in the brain, microglia can act as a sentinel for neurons [31]. Thus, these results suggest that CX3CL1 activity may result from the direct activation of signaling pathways in microglia. CX3CL1 can prevent activated microglia from producing pro-inflammatory cytokines and reactive oxygen species, release neuroprotective adenosine to stimulate astrocytes to upregulate glutamate transporter expression and modulate the entry of hematogenous leukocytes into the brain [32] As already discussed in the previous paragraph, the activity of CX3CL1 is influenced by metalloproteases, such as ADAM10 and ADAM17. Therefore, these enzymes also play a key role in neuroinflammatory processes. ADAM-10 and -17 are present in various regions of rodent brains by Northern blot analysis, and immunohistochemical and in situ hybridization studies showed that they are expressed in endothelial cells and astrocytes. The expression of ADAM17 has been observed to increase during the inflammatory response. ADAM17 participates in neuroinflammation as it is reported to be responsible for the proteolytic activation of the membrane precursor of TNFα, a cytokine critically involved in inflammation and with relevance in AD. ADAM17 also targets many receptors, as cleavage of this type of molecule offers an additional way to regulate the cell’s response to cytokines and chemokines. For example, the IL-1 (IL-1R) and IL-6R receptors are processed by ADAM17. In addition to cytokines and chemokines, many soluble growth factors associated with inflammation are cleaved by ADAM17 for further maturation and secretion. TGFα is one of the growth factors confirmed to be cleaved by ADAM17 as a byproduct of TNFα research. ADAM17 has also involved in EGF signaling, as ADAM17 deficient mice need the phenotype of mice with an EGF signaling defect, such as prenatal lethality accompanied by severe developmental defects of the heart [33]. Neuroregulins (NRGs) are also substrates of ADAM17 and ADAM10: it has been reported that ADAM17 can cleave NRG1 type III to negatively control myelination of the peripheral nervous system and ADAM10 regulates the formation and maintenance of synapses by cleaving several key localized synaptic proteins both in the pre-synapse and the post-synapse, for example, N-cadherin, neuroligin-1 (NLGN1) and neurexin (NRXN). Both ADAM10 and ADAM17 are involved in the development of the nervous system through the activation of neural cell adhesion and the growth of neurites by the cleavage of L1 and the adhesion molecule of neuronal cells. These ADAMs act on a large group of CAMs, thus regulating cell–cell interactions, and it has been shown that the elimination of some cell adhesion proteins is relevant in signal transduction. In the CNS, they split different cell adhesion molecules, for example, EphrinA5, NCAM, NrCAM, and Nrp1, and, thus, regulate the targeting of axons and the growth of neurites. In PNS, ADAM10 is the main protease to release the soluble ectodomain DR6 (sDR6), which in turn negatively regulates axonal myelination by Schwann cells (SC) [33,34].

## 4. CX3CL1 and ADAMs/MMPs in Neurodegenerative Disease

Neuroinflammation was initially thought to be a consequence of neurodegenerative diseases, but more recently, it has been revealed that neuroinflammation itself plays a significant role in the development and progression of the disease, including Alzheimer’s disease (AD), Parkinson’s disease (PD), Huntington’s disease (HD), and other neurodegenerative diseases. Therefore, neuroinflammation is seen as a therapeutic target for neurodegeneration. Neuroinflammation can be modulated by glial-neuron signaling through the CX3CL1/CX3CR1 pathway. The predominant function of CX3CL1 within the CNS is believed to be to reduce the pro-inflammatory response, and many studies have shown neuroprotective effects. However, in some cases, CX3CL1 appears to promote neurodegeneration [35]. This dual effect is dependent on the form of CX3CL1; the neuroprotective capacity of CX3CL1 lies in its soluble isoform in AD and Parkinson’s disease [36,37]. Therefore, perhaps only the soluble subtype of CX3CL1 can access the cytoplasm after binding with CX3CR1 and consequently trigger the downstream signaling pathways. Even by modulating the activity of ADAMs, it is possible to regulate neuroinflammation and the progression of neurodegenerative diseases. For example, ADAM10 has been shown to cleave APP within the Aβ sequence, which precludes the formation of pathological Aβ plaques in AD. ADAM10 also plays a dual role in prion disease since, on the one hand, ADAM10 inhibits the transition of cellular prion protein (PrPc) to pathological prion protein (PrPsc), but on the other hand, promotes the spread of PrPsc [34]. ADAM17-dependent TNFα cleavage has been shown to inhibit neuronal apoptosis through NFκB activation in rat mixed cortical cultures. Others have shown that ADAM17 can contribute to brain repair by inducing the proliferation and migration of neuronal stem cells in interaction with the endocannabinoid system [33].

### 4.1. CX3CL1 and ADAMs/MMPs in Alzheimer’s Disease

Alzheimer’s disease (AD) is the most common form of dementia affecting nearly 45 million people worldwide. The main neuropathological features of AD are the accumulations of β-amyloid plaques, tau tangles, neuroinflammation, and synaptic and neuronal loss, the latter being the most vital correlating factor with memory and cognitive impairment. Many of these pathological signs influence each other during the onset and progression of the disease. Recent genetic evidence suggests the possibility of a causal link between impaired immune pathways and synaptic dysfunction in AD. Emerging studies also suggest that immune-mediated synaptic pruning could initiate the early-stage pathogenesis of AD [38]. During AD, CX3CL1/CX3CR1 signaling can control disease progression by inhibiting inflammation and tau phosphorylation, but at the cost of increasing deposition of β-amyloid fragments. It has been suggested that, at the onset of AD, intra-neuronal accumulation of β-amyloid causes a slight decrease in CX3CL1/CX3CR1 signaling, resulting in increased Aβ phagocytosis and hyperphosphorylation of tau [39]. However, CX3CL1-mediated neuron-glia crosstalk in the context of AD has reported conflicting results [40,41]. The CX3CL1/CX3CR1 interaction plays a vital role in maintaining a healthy and anti-inflammatory condition in the brain. Some transcription factors (P38, β-catenin, NF-κB) and associated molecules (AKT and GSK3-β) have been linked to AD pathology and the pathway CX3CL1/CX3CR1. In AD, the presence of β-amyloid plaques causes CX3CL1 to no longer bind its receptor; this causes hyperactivation of the microglia with consequent activation of p38, which has been seen to have increased its expression in the early stages of AD. So β-amyloid plaques have been found to stimulate microglia to rapid activation of MAPK p38 resulting in the upregulation of proinflammatory cytokines, such as IL-1 and TNF-α. Furthermore, IL-1 activates MAPK p38 in astrocytes and neurons, causing excessive inflammation and phosphorylation of tau [42]. The peptides have also been reported to stimulate microglia to express high levels of IL-1, IL-6, and IL-18. Proinflammatory cytokines’ production induces astrocytes’ activation, followed by microgliosis [43]. CX3CL1 is also involved in the activation of the NF-κB pathway [44]. NF-κB plays a vital role in the inflammation, signaling, and progression of AD. Activation of NF-κB contributes to increasing β-secretase in neuronal cells in vitro and in vivo. Diseased AD brains reduce increased levels of BACE1 and NF-κB p65, and NF-κB p65 leads to increased activity of the BACE1 promoter and BACE1 transcription, while knockout of NF-κB p65 reduces gene expression BACE1 in cells [45]. In addition to having binding sites in the promoter region of the genes involved in amyloidogenesis, NF-κB also has binding sites in the promoter region of the genes involved in inflammation [46]. NF-κB regulates the expression of pro-inflammatory cytokines, such as IL-6 e IL-1β and TNF-α, which are elevated in the brains of AD patients. These cytokines (IL-6 and IL-1β) not only increase inflammation but also induce brain cell death by apoptosis. In the brains of AD patients, activated NF-κB was found predominantly in neurons and glial cells in the areas surrounding the Aβ plaques [45]. It has been shown that in people with Alzheimer’s, there is a significant increase in CSF (cerebrospinal fluid) and blood concentrations of soluble CX3CL1. The elevated levels of soluble CX3CL1 in AD patients can be attributed to the role it plays as an inflammatory mediator. In addition, a disease stage-dependent difference in sCX3CL1 concentration was observed. The concentration of sCX3CL1 increased in the early stages of AD and then gradually decreased in the development and progression of the disease. This may be associated with the activation of the acute inflammatory process in the early stages of the disease [47]. It is, therefore, possible that neuroinflammation triggers a protective mechanism through the increase in the activity of proteases that cut CX3CL1 by releasing its CSF and blood soluble form, which results in increased levels of sCX3CL1. Fan et al. demonstrated that the sCX3CL1 fragment generated by protease cleavage affects the deposition of β-amyloid fragments and found that a higher expression of sCX3CL1 was sufficient to reduce the deposition of β-amyloids in large regions of the brain since they have a whole structure of the total levels of APP determines-C99 and C83, products of APP at length cleaved by BACE1 and α-secretase. The reduction of both C-li fragments of APP would be related to the termination of the amyloid deposition [12]. Among other things, the soluble form of CX3CL1 and the deposition of β-amyloids also play a role in the tau that characterizes Alzheimer’s. It has been shown that CX3CL1 reduces tau pathology levels and forms a protective effect against the loss of neurons. It was also noted that there is no statistically significant reduction in total tau, only in phosphorylated tau levels [16]. Furthermore, a reduction in the active form of GSK3-β porcelain, responsible for the phosphorylation of tau, was also observed [48]. The membrane form of CX3CL1 also influences tau disease. It has been seen that the tau protein binds the same membrane receptor as CX3CL1 or CX3CR1 and that these two proteins compete for binding. Marta Bolós et al. used affinity photography for Tau chromatography retained in the column in the absence or presence of CX3CL1 and observed that the binding of Tau to CX3CR1 was weaker in the presence of CX3CL1. It was also evaluated whether competition between Tau and CX3CL1 affected the internalization of Tau by microglia. The results showed that in the presence of CX3CL1, the internalization of Tau is reduced [40]. Therefore, on the one hand, fraktaline in its soluble form has a beneficial effect on pathology and, therefore, on AD; on the other hand, it’s membrane forms tau phagocytosis because it competes with its binding to the CX3CR1 receptor of microglia. Therefore, CX3CL1 could represent a fundamental element for the slowdown of Alzheimer’s disease and a potential target for therapeutic intervention.

ADAMs also play an important role in Alzheimer’s disease. Both ADAM17 and ADAM10 act as relevant alpha-secretases to cleave the amyloid precursor protein APP and produce a soluble and non-amyloidogenic fragment, APPsα, thus having a neuroprotective effect precluding the formation of pathogenic Aβ peptides and for AD. ADAM10 acts as neuronal α-secretase under non-stimulated constitutive conditions, while ADAM17 is known to be the “stimulated α-secretase” and responds to stimuli, such as phorbol esters and activation of the muscarinic M1 receptor, thus carrying out a fundamental role in the pathophysiology of AD [49]. Interestingly, ADAM17 has been shown to localize in brain areas that contain amyloid plaques [50]. It has recently been reported that a variant of ADAM17 leading to the loss of its function is associated with the pathogenesis of AD in humans [51]. Mutations in the ADAM10 prodomain were also associated with late-onset AD [52]. Overexpression of these mutated forms of human ADAM10 in a mouse model of AD showed an increase in amyloid plaques and reactive gliosis compared to overexpression of wild-type human ADAM10, indicating that the mutated form of ADAM10 has reduced α-secretase activity towards APP. In contrast, moderate neuronal overexpression of wild-type ADAM10 in an AD mouse model showed increased sAPPα secretion, reduced Aβ peptide and plaque formation, and alleviated impaired long-term potentiation and cognitive deficits, indicating that activation of ADAM10 could perform a therapeutic activity for AD [53]. The activity of ADAM17 can be ambivalent depending on the cellular target on which it acts. It is interesting to note the fact that neuronal ADAM17 can have protective effects on the development of AD by triggering the non-amyloidogenic pathway, and microglial ADAM17 can have the opposite effects. This is demonstrated by the substrate of ADAM 17 TNFα, which is involved in the pathogenesis of AD through pathways that induce neuroinflammation. As previously mentioned, the deposition of amyloid plaques induces the microglia to release pro-inflammatory cytokines, including TNFα, which induce neuronal death. Ablation of TNFα and its target receptor TNFR1 in a mouse model of AD lowered cognitive decline by preventing disease-associated learning and memory deficits [51,54]. Furthermore, ADAM17 may contribute to the elimination of the triggering receptor expressed in myeloid cells (TREM2). Trem2 is a genetic risk factor for AD and an important regulator of the neuroinflammatory component of the disease, and a reduction in its spread may be useful [55]. Regulators of ADAMs also play a fundamental role in the pathogenesis of AD, as the pro-inflammatory cytokine interleukin-1 can increase the activity of ADAM-17 and the formation of soluble APP in astrocytes and, at the same time, decrease the production of Aβ. Interestingly, interleukin-1 can also stimulate APP translation in cells [50]. Another fundamental secretase in AD is γ-secretase. Since γ-secretase mediates the final cleavage that releases Aβ, γ-secretase has been extensively studied as a potential drug target for the treatment of AD. However, γ-secretase inhibitors have been shown to cause side effects in clinical studies due to the inhibition of Notch signaling. Therefore, modulation, rather than inhibition of γ-secretase, is preferred to alter the production of Aβ in a therapeutically relevant way without interfering with essential cellular processes. Among the inhibitors of γ-secretase are the JLK compounds of isocoumarin, which block the production of Aβ at the level of the γ-secretase but do not affect Notch processing. Some non-steroidal anti-inflammatory drugs (NSAIDs; for example, ibuprofen, indomethacin, and sulindac sulfide) can reduce the production of the highly aggregated Aβ42 peptide. Enzyme kinetic studies and displacement experiments show that selective NSAIDs interact with γ-secretase at a site distinct from the active site. The cleavage site within the Notch transmembrane domain is similarly affected, but this subtle change does not inhibit the release of the intracellular domain and therefore does not affect Notch signaling. For this reason, these agents may be safer as therapeutics for Alzheimer’s than inhibitors that block the active site or docking site. In addition, ATP and other nucleotides were also tested for effects on γ-secretase and were found to selectively enhance the proteolytic processing of a purified recombinant APP-based substrate without affecting the proteolysis of a Notch counterpart. Additionally, some compounds known to interact with ATP binding sites have been found to selectively inhibit APP processing compared to Notch.

### 4.2. CX3CL1 and ADAMs/MMPs in Parkinson’s Disease

Parkinson’s disease (PD) is a progressive neurodegenerative disease that occurs mainly in adults aged 60 and over and is more prevalent in men than in women. It is the second most common neurodegenerative disorder and affects at least 6 million people worldwide [56]. The pathophysiology of Parkinson’s disease is characterized by aberrant aggregation of α-synuclein and its accumulation in Lewy bodies, production of mitochondria, lysosomes, or vesicle transport, stable striatal dopamine, and a progressive lack of dopaminergic neurons within the substantia nigra pars compacta, from synaptic transport problems and neuroinflammation. These pathological mechanisms collectively result in accelerated neuronal death of neurons, but neuropathology involves many other motor and non-motor circuits. In addition to the dopaminergic system, other neurological systems are also involved in PD, including the glutaminergic, cholinergic, noradrenergic, and GABAergic pathways [56,57]. Neuroinflammation is also a major cause of neurodegenerative changes in the brains of PD patients, and this is activated by immune and microglial cells. In fact, in Parkinson’s patients, an increase in the expression of inflammatory chemokines/cytokines and the activation of the nuclear transcription factor NF-κB, which controls the target genes coding for proinflammatory cytokines, and adhesion molecules, chemokines, was detected as growth factors and inducible enzymes [58]. CX3CL1/CX3CR1 signaling exerts a relevant effect on PD-related inflammation and neurodegeneration. Indeed, it has been shown that fraktaline regulates microglial IL-1β expression via AKT and the PI3-kinase pathway, induces translocation of the p65 subunit of NF-κB into the nucleus via AKT, and activates the transcription factor CREB in neurons of the hippocampus. The JAK-STAT pathway was shown to play a significant role in PD, as its inhibition can protect itself from dopaminergic degeneration and neuroinflammation in a model that overexpresses α-synuclein. It has been indicated that the imbalance between the p38 MAPK and PI3K/AKT cascades plays a critical role in the pathogenesis of PD. In particular, p38 MAPK activation generally improves dopaminergic cell apoptosis, ER stress, microglial activation, and oxidative and mitochondrial temperature, while the PI3K/AKT axis can be quite neuroprotective through several mechanisms, including the inactivation of GSK-3β also implicated in dopaminergic neurodegeneration. NF-κB and its upregulation were also observed in patients with PD, along with an increase in the expression of TNF-α, IL-1β, and IL-6. The use of NF-κB inhibitors has been shown to inhibit neuroinflammation and dopaminergic degeneration in mouse models of PD [59]. Deletion of the fractalkine or CX3CR1 receptor has been shown to increase dopaminergic neuronal loss and increase microglial activation. As in AD and also in PD, the beneficial effect of CX3CL1 on the disease depends on its form, soluble or membrane. In CX3CL1 knockout mouse models in which Parkinson’s was induced, only the soluble form of fraktaline was shown to inhibit neurotoxicity, accompanied by improved motor impairment, prevention of dopaminergic neuronal loss, and reduced microglial activation in animals. Functionally, soluble fraktaline has been shown to inhibit microglial activation, suppressing the release of pro-inflammatory cytokines, including TNF-α and IL-1β, as well as the expression of CD68 and CD11b in mice with PD [35,59]. Furthermore, microglial inhibition by CX3CL1 was also demonstrated through the presence of the major histocompatibility complex II (MHCII), whose expression was reduced. Activation of microglia induces expression of MHCII and indicates a harmful form of activation of microglia [36]. Furthermore, the soluble form of CX3CL1 has been shown to have a neuroprotective effect against α-synuclein-mediated damage by reducing the loss of dopaminergic neurons by 40–50%. In contrast, the native form and membrane form of CX3CL1 showed no effect. This could suggest that the membrane and secreted forms of FKN have alternative functions in the CNS. A possible explanation could be altered receptor signaling with sFKN because the secreted form allows for receptor internalization [60]. The internalization of CX3CR1 in the microglial cytoplasm following the binding with the soluble fraktaline is responsible, in fact, for the activation of the downstream signaling. CX3CR1-deficient mice have been shown dopamine-deficient production compared to CX3CR1 wild-type mice, showing dopamine injection a significant role in dopamine signaling level in physiological conditions free of toxins, as well as the importance of basic physiological levels of CX3CR1 in dopaminergic function [61]. Given the important role of CX3CL1 in PD, clinical trials on PD patients were conducted to analyze the levels of fraktaline in their serum. CX3CL1 levels are markedly elevated in the serum of PD patients compared to the serum of non-PD control patients, but CX3CL1 levels tend to decrease as the disease progresses [62]. It was shown that the gradual change in fraktaline levels was related to the increase in motor aberrations in these patients. The results demonstrated that CX3CL1 plays a key role in the early course of Parkinson’s disease. Serum fraktaline is increased in the early stages of PD when motor disability is mild, but as the disease progresses, levels of this cytokine gradually decrease. Significantly, the declining levels correlated with progressive motor disability in patients with PD [63]. This may be because CX3CL1 levels are increased in the early stages of PD as a reflection of a protective anti-inflammatory effect, so there is a high trend in the early stages of PD and progression as the disease progresses. Therefore, CX3CL1 could represent a new element for the early diagnosis of PD.

Even in Parkinson’s disease, ADAMs have a fundamental role in the pathogenesis of the disease. Several studies have reduced the ADAM10 gene, and it has been suggested that it is involved in the development of neurogenerative disorders as its activity is involved in APP, which plays an important role in motor temperature and cognitive decline in PD. In 2021 Miaomiao Zhou et al. studied the rs514049 polymorphism of the ADAM10 gene (located in the promoter region, thus affecting its expression) and ADAM10 plasma levels in patients with PD and healthy controls from a Han Chinese population. The rs514049 CC genotype was seen to be yes while associated with an increased risk of PD, particularly in Chinese male patients with PD, compared to differences between female patients with PD. Sex is known to be one of the most important risk factors for PD; in fact, the number of male patients with PD is 1.5 times greater than the number of females. This suggests that the rs514049-C allele may be involved in the risk of PD via a gender-dependent mechanism. One explanation of the possible CC outcome does not have the risk that the system in female patients with PD could be that estrogen activates ADAM10 to exert a neuroprotective function on dopaminergic neurons by cleaving the cytokines involved in neuroinflammation for the pathogenesis of PD. Furthermore, they found sufficient plasma ADAM10 levels in Chinese patients with PD, indicating that plasma ADAM10 probably participates in the development of PD. The size of ADAM10, in addition to being involved in the processing of APP, caused changes in the morphology and functions of the microglia, altering the microglia synapses. This metalloprotease is also involved in several inflammatory pathways. Therefore, the activity of ADAM10 in PD is involved in the accumulation of Aβ, synaptic dysfunction, and neuroinflammation. This suggests that ADAM10 plays a fundamental role in PD, and its regulation could be useful in the early diagnosis and therapy of PD [61]; ADAM17 is also important in PD. In 2019 Wei-Wei Li et al. selected an SNP tag of the ADAM17 gene for its function of encoding TACE, the p75ECD sheddase that has been found important in modulating p75NTR-induced neurotoxicity in neurodegenerative diseases [64]. p75NTR also acts as a negative modulating factor for cell survival and mitochondrial insult sensitivity of mesencephalic dopaminergic neurons, and altered levels of p75NTR expression may contribute to the pathogenesis of PD. The expression level of p75NTR is also upregulated in dopamine neurons of animal models with PD. Therefore, a malfunction of ADAM17 involves an upregulation of p75NTR which can mediate cell death and neurite degeneration of dopaminergic neurons [65]. BACE1 is also a secretase involved in PD as this enzyme generates the toxic A*β* peptides found in PD. Indeed, a BACE1 SNP has been associated with an increased risk of PD [66].

### 4.3. CX3CL1 and ADAMs/MMPs in Multiple Sclerosis (SM)

Multiple sclerosis (MS) is defined as a chronic inflammatory disease of the central nervous system (CNS). The disease almost always begins before the age of 40, affects about three females to one male, and is a leading cause of disability [67]. MS leads to large focal lesions in the white matter of the brain and spinal cord and is characterized by primary demyelination with variable axonal loss. Indeed, it later became clear that the lesions are also present in the gray matter, including the cortex, basal ganglia, brainstem, and gray matter of the spinal cord. The lesions occur in conjunction with an inflammatory reaction consisting of T lymphocytes, B lymphocytes, and plasma cells that begin around the veins and post-capillary veins. Thus, in the early stages of lesions, perivenous demyelination is observed, and these lesions coalesce into confluent demyelinated plaques. Similar perivenous and confluent demyelinated lesions also form in gray matter, including the cerebral and cerebellar cortex, deep brainstem nuclei, and gray matter of the spinal cord. The inflammatory state around and within the lesions is present in all stages, both early and progressive [68,69]. Thus, neurodegeneration in MS is influenced by demyelination which is a slow process initiated by acute lymphocytic inflammation and subsequently driven by diffuse chronic myeloid parenchymal and meningeal lymphocytic inflammation. In MS, oxidative stress, mitochondrial damage, and subsequent ion channel dysfunction also appear to impact neurons and axons. Indeed, several ion channels show compensatory changes in response to the inflammatory stimulus by altering their relative distribution in the neuron, a process that eventually becomes maladaptive and perpetuates neuroaxonal damage [67]. As in previous neurodegenerative diseases, also in MS, the inflammatory cytokine CX3CL1 plays a fundamental role in the pathogenesis of the disease. In 2003 Sandra Hulsshof et al. hypothesized that CX3CL1 might influence tissue remodeling by MAP kinase activation; such remodeling is a pivotal event in the development of MS lesions. In their study, they demonstrated a functional response to CX3CR1 signaling in microglial cells, which led them to suggest that CX3CL1 might be involved in tissue remodeling during the development of MS lesions. This suggestion is strengthened by other studies where CX3CL1 of neuronal origin was shown to induce proliferation, activation, and/or migration of microglia into damaged brain sites [70]. In 2005 Dan Sunnemark et al. demonstrated that the chemokine CX3CL1 is expressed and not regulated within inflammatory CNS lesions in rats with experimental autoimmune encephalomyelitis (EAE), a rodent model of MS. Furthermore, they demonstrated that cells expressing the CX3CR1 receptor, mostly macrophages and/or microglia, accumulate densely within inflammatory lesions. This is consistent with the hypothesis of a role of CX3CL1/CX3CR1 in the local control of leukocyte infiltration in CNS lesions in EAE rats and, thus, also in MS., They hypothesized that in MS patients, it is likely that CX3CL1 is cleaved and released from the neuronal membrane by metalloproteases, and this would create a chemoattractant gradient which, in concert with other locally active proinflammatory mediators, could accommodate the extensive accumulation and activation of microglia within the neuronal membrane, inside damaged brain sites [71]. Furthermore, the soluble form of CX3CL1 may also contribute to the infiltration of leukocytes expressing CX3CR1, including macrophages and true T and NK cells. These cell types have crucial roles in EAE as mediators of tissue destruction and/or disease regulation. This was demonstrated in 2015 by Kevin Blauth et al., who considered the role of CX3CL1 in the development of the inflammatory response. They found that CX3CR1 is a marker for CD4+CD28− T lymphocytes, which are expanded in MS patients. CD4+CD28− cells preferentially migrate towards a CX3CL1 gradient in vitro, suggesting that CX3CR1+CD4+ cells identified in MS lesions are attracted to the CNS due to CX3CL1 chemotaxis. These types of cells were found in abundance in the CSF of patients with RRMS (Relapsing-Remitting Multiple Sclerosis) compared to control patients. Furthermore, they found that secreted CX3CL1 is a key regulator of ICAM-1 expression, which in turn influences inflammatory cytokine production. In vitro studies on the mechanisms of CX3CL1-induced migration revealed that CX3CL1 induces increased IFN-γ and TNF-α gene expressions and increased IFN-γ secretion in CD4+ T lymphocytes derived from RRMS patients but not in control patients. This led them to conclude that increased CX3CL1 concentrations in cerebrospinal fluid and serum may contribute to the induction of proinflammatory cytokines in RRMS [72]. Furthermore, Bieke Broux et al. found that fractalkine is upregulated in the cerebrospinal fluid of MS patients already after the first clinical manifestation and in the brain lesions of chronic MS patients. Since fractalkine is already upregulated in the early stages of MS patients, these cells (CD4+CD28− T lymphocytes) have the potential to migrate into the brain very early in the disease course. Once in the brain, these cells could release cytotoxic mediators in response to T cell receptor (TCR) reaction by MS-related autoantigens or in an unrestricted manner by MHC via NK-like killing mechanisms. This report provides evidence that CD4+CD28− T lymphocytes play a part in the initial disease-related inflammatory processes in the brain of a subset of MS patients [73]. Jeffrey H Mills et al. also demonstrated the role of CX3CL1 in lymphocyte migration into the CNS using extracellular adenosine-induced FKN at the choroid plexus [74]. The increased expression of CX3CL1 in CSF and serum of MS patients, the enrichment of CX3CR1+CD4+ cells within CSF and MS lesions, and the proinflammatory effect of fractalkine on cytokine secretion and upregulation of the adhesion molecule suggest that its induction of inflammatory cell migration may represent an attractive therapeutic target. The CX3CL1/CX3CR1 axis may therefore represent a selective therapeutic target of MS. This hypothesis was supported using a potent and selective CX3CR1 inhibitor, AZD8797, in the EAE model introduced by MOG1-125 in rats. Initiation of AZD8797 treatment before or after disease onset reduced the clinical symptoms and pathological signs of EAE in a concentration-dependent manner. The use of AZD8797 is supposed to block the infiltration of CX3CR1-expressing cells to the CNS from the periphery that no longer sense CX3CL1, and as a result, this indirectly affects demyelination, neuronal pathology, microglial activation, and other inflammatory processes in the CNS [75]. A conflicting result was shown by Lampron et al. in 2015, in which deletion of CX3CR1 blocked the clearance of myelin debris by microglia, which greatly affected the integrity of axons and myelin sheaths, preventing proper remyelination [76]. These results highlight the crucial role played by CX3CR1 in myelin clearance and show that there can be no efficient remyelination following primary demyelinating lesions if myelin clearance by microglia is impaired. Furthermore, CX3CL1 and CX3CR1 have established factors in the modulation of pain perception through a central proalgesic mechanism. NPP (neuropathic pain) is the second worst symptom induced by MS and has also been reported in MS patients before the time of diagnosis and, thus, may be a promising pre-diagnostic marker to facilitate early diagnosis of the disease. CX3CL1 expression is increased in the dorsal root ganglion (DRG) and SC during the initial inflammatory phase of EAE induction before tissue damage of demyelination. Consequently, this confirms the importance of the immune system in pain induction before any detectable tissue damage or injury. Thus, CX3CL1 is a nociceptive mediator involved in the early induction of NPP induced in the early stages of inflammation by the immune system before any detection of myelin damage or injury. CX3CL1 signaling in dorsal horn SC neurons activates ascending pathways involved in nociceptive transmission. Therefore CX3CL1 and its receptor CX3CR1 could be evaluated as biomarkers of MS-induced NPP [77]. Based on the role of inflammation in MS, it has been shown that a large part of MMPs is involved in neuroinflammatory disorders. For example, ADAM17 expression was associated with blood vessel endothelium, activated macrophages/microglia, and parenchymal astrocytes in MS white matter. Increased levels of ADAM17 immunoreactivity have been visualized in active lesions with myelin rupture; this certainly implies a role in the disease process. ADAM17 has been shown to mediate the cleavage and detachment of the adhesion molecules L-selectin and VCAM-1. Increased vascular surface adhesion molecules were shown on the cell surface of vascular endothelial cells compared with normal-appearing white matter (NAWM) and white matter controls in the MS lesion. Thus, increased ADAM17 activity leads to an increase in the number of membrane-bound adhesion molecules available for leukocyte infiltration into MS lesions [78]. Furthermore, ADAM17 could also be detected in the cerebrospinal fluid of MS patients but not in non-inflammatory controls. This finding suggests that immune cells in the MS lesion prolong the ongoing inflammatory reaction by releasing ADAM17. ADAM17, in turn, once released, processes membrane-bound TNF-α by activating it. TNF-α is expressed by astrocytes and macrophages in MS lesions; consequently, ADAM-17 also appears to play a substantial role in the pathogenesis of MS. In addition to pro-TNF-α/ADAM17 cleaving ectodomains of other receptors and ligands come TNFR-2 and L-selectin. Changes in serum TNFR-2 levels have been shown to correlate with disease activity in MS [79]. In MS, the balance between ADAM17 and its inhibitor TIMP3 is deregulated. Indeed, it was demonstrated that ADAM17 mRNA was increased at the peak of the disease, while TIMP3 mRNA was decreased [78]. Thus, the use of inhibitors against MMPs could be a method to reduce neuro and brain damage in MS, the use of MMP inhibitors, such as hydroxamic acid compounds, minocycline (tetracyclic antibiotic), interferon-β, has been shown to reduce the ability of T cells to migrate into the CNS [80,81]. Besides ADAM17, other MMPs are deregulated in MS. For example, MMP-9 and MMP-7 are upregulated in multiple sclerosis plaques compared to healthy controls. Furthermore, strong MMP-9 immunoreactivity was seen in the blood vessel walls of the MS lesions by immunohistochemical staining, whereas no significant MMP-9 immunoreactivity was found in the normal white matter of the control cases. An analysis of the transcriptional expression of MMP-9 in CSF cells from lumbar punctures performed during relapses and clinically stable phases of MS was also performed. MMP-9 transcripts were undetectable in control CSF, while all multiple sclerosis samples were positive with similar levels during relapses versus clinically stable phases of the disease [82].

### 4.4. CX3CL1 and MMPs in Huntington’s Disease

Huntington’s disease (HD) is an autosomal dominant neurodegenerative disease with an onset of symptoms in the 30 and 40 years of age. About 5% of cases have onset in childhood or adolescence, referred to as juvenile HD. Conversely, there are clinical cases in which patients remain asymptomatic until the seventh or eighth decade of life, known as late-onset HD. The duration of the disease from diagnosis to death is approximately 15–20 years. The primary site of pathology is the neostriatum, which includes the caudate nucleus and putamen. The disease is caused by an expansion in the number of CAG repeats in the huntingtin gene on chromosome 4, leading to an elongation of the polyglutamine tract within the gene product called “huntingtin”; therefore, HD is considered one of the so-called polyglutamine diseases. The mutant huntingtin protein has a toxic function that causes neuronal death. The most prevalent brain atrophy is caused by the loss of structural and functional connectivity between the striatum and other parts of the brain. Clinically, HD is characterized by motor, cognitive and psychiatric symptoms due to atrophy of the basal ganglia (especially neostriatum) and cerebral cortex. Motor symptoms, such as chorea, dystonia, and bradykinesia, consist of rapid or slow involuntary movements of the face, trunk, and limbs or even slower movement and stiffness of the limbs. Patients with HD also have reduced verbal fluency and memory difficulties. Additionally, psychiatric symptoms of the disease include apathy, anxiety, irritability, depression, obsessive-compulsive behavior, and psychosis. Psychiatric disorders are also common many years before the onset of symptoms in the pre-manifest phase [83,84]. At the molecular level, however, mutant huntingtin causes neuronal dysfunction and death through some mechanisms, such as proteolytic stress, defects in transcription and translation, mitochondrial dysfunction and disorders in energy metabolism, disorders in the cytoskeleton, and axonal transport. The number of CAG repeats in the mutant allele has also been shown to be inversely correlated with age at disease onset and age at death. The number of CAG repeats is directly correlated with an HD progression rate [85]. Although there are still few studies on it, it seems that CX3CL1 and its receptor may also play a role in the pathogenesis of HD. Interestingly, network analysis of microarray data from postmortem human tissue revealed that CX3CL1 is an important novel factor in the pathogenesis and survival of HD [86]. Furthermore, given that microglial cell activation is an early event, it is thought that the CX3CL1/CX3CR1 axis may contribute to the disease process leading to significant inflammation in the later stages of the disease [87]. The analysis of the CX3CL1/CX3CR1 axis was investigated by the use of male and female R6/1 transgenic mice expressing human mHtt exon-1 containing 115 CAG repeats of different ages. Following qRT-PCR analysis, no change in CX3CR1 gene expression was found in the striatum of R6/1 mice compared to WT animals at any age studied, whereas a reduction in CX3CL1 gene expression was observed in the striatum of R6/1 mice in both at 8 weeks, before disease symptoms manifested, compared to WT, and also at later ages. A comparison was also made of CXC3CL1 protein levels between R6/1 and WT mice, and significant evidence of striatal FKN concentration was observed in R6/1 mice at 6, 8, 12, and 20 weeks but not at 30 weeks of age [88].

Proteolytic processing of mHTT produces toxic fragments, which cause neurotoxicity and neuronal death predominantly in the striatum and cortex; thus, mHTT proteolysis contributes to the pathogenesis of HD [89]. Matrix metalloproteinases (MMPs) are modifiers of proteolysis and mHTT toxicity. Swati Naphad et al. found altered expression of MMP-2 and MMP-9 (gelatinase), MMP3/10, and MMP14 activity in HD model cells compared to control cells. MMP-3/10 levels were found to be elevated in HD cells compared to the control. This was confirmed by immunofluorescence analysis which also revealed an altered localization of MMP-3/10 in which, in control cells, they were predominantly expressed in the cytoplasm, while in HD cells, nuclear expression is also observed, which appears to be also associated with apoptotic cells. Western blot analysis and RT-PCR analysis showed that MMP-14 levels were decreased in HD cells compared to control, and MMP-14 expression was found to be strongly nuclear in HD cells, which may associate with nuclear HTT immunoreactivity. A higher expression of pro-MMP was detected in HD cells compared to control, and immunofluorescence analysis again detected nuclear localization of MMP-9 in HD cells compared to cytoplasmic localization in control cells. As regards MMP-2, on the other hand, upregulation of its expression was observed in association with apoptotic cells, thus suggesting a role in cellular apoptosis. Dysregulation of MMP activity is accompanied by concomitant changes in the levels of the endogenous MMP inhibitor, TIMP. Indeed, reduced levels of TIMP-1 and TIMP-2 were observed in HD cells, suggesting that part of the altered expression and activity of MMPs is due to the lower abundance of these endogenous inhibitors. Immunofluorescence analysis revealed increased detection of MMP/TIMP in the nucleus or aggregates of HD cells, suggesting a potential interaction with that [84]. The fact that MMPs have a distinctive role in HTT cleavage was also confirmed by Miller et al., who performed an RNAi screen to identify proteases that, when knocked down, reduced the production of toxic N-terminal HTT proteolytic fragments. These demonstrated that the knockdown of MMP10, MMP-14, and MMP-23 in cultured striatal cells expressing mutant HTT reduced toxicity. Furthermore, MMP10 can directly cleave HTT, and the production of toxic fragments of HTT is reduced after the silencing of MMP-10. Other evidence for the involvement of MMPs in HD comes from an animal model with a 3-mitropropionic acid-induced disease that mimics the brain lesions seen in Huntington’s disease [90]. J. Duran-Vilaregut et al. investigated the role of MMP-2 and MMP-9 in blood–brain barrier (BBB) degradation in striatal lesions. MMP-9 was present in the majority of degraded blood vessels in the damaged striatum, while it was absent in healthy tissue vessels. In the same animals, MMP-2 staining was just detected near degraded blood vessels. MMP-9 immunostaining studies, in situ zymography, and MMP-9 inhibitor studies confirmed that the protease activity detected in degraded striatal blood vessels could be almost exclusively attributed to the active form of MMP-9 [91]. Another study reported a direct correlation between increased levels of MMP-3 and MMP-9 in the cerebrospinal fluid (CSF) of HD patients and disease worsening in such patients. So these data also imply that MMPs are modulators of HD neuroinflammation.

## 5. Conclusions

CX3CL1 role in neurodegenerative diseases has shown conflicting results based on the different forms of CX3CL1 and on the characteristics of the neurodegenerative disease types. For example, in Parkinson’s pathology, the soluble form of CX3CL1 has been shown to have a neuroprotective effect against α-synuclein-mediated injury by reducing dopaminergic neuron loss by 40-50%; differently, the membrane form has not a similar effect. Likewise, also in Alzheimer’s disease, the sCX3CL1 form has beneficial effects, influencing the β-amyloid fragments deposition in large regions of the brain. Contrariwise, the CX3CL1 membrane form in Alzheimer’s disease has a negative effect on preventing tau phagocytosis by its competition in binding the microglia receptor CX3CR1.

From the published results, however, the action mechanisms of CX3CL1 are not yet fully understood; various evidence underlines its fundamental role in the neuroinflammatory process occurring in neurodegenerative pathologies. In fact, an increase in CX3CL1 expression has been highlighted in the CSF and serum of patients with multiple sclerosis, Alzheimer’s, and Parkinson’s, already in the earliest stages of the disease. In contrast, in Huntington’s disease, there was a low expression in protein and gene coding CX3CL1 (Figure 3).

From what has been observed from the reported literature, several questions arise: (i) is CX3CL1 the cause or consequence of the neuroinflammatory process in neurodegenerative diseases? (ii) Furthermore, what is the form of CX3CL1 most involved in the neuroinflammatory state from which it can no longer be regressed? And consequently, what are actually the activated and/or inhibited elements that lead to the different conditions? To answer these questions, there are still little supporting data, but we are certainly moving in the right direction.

Not least, could CX3CL1 represent an early-stage marker for the diagnosis of these neurodegenerative pathologies? And consequently, a common therapeutic target for neurodegenerative pathologies?

## Figures and Tables

**Figure 1 ijms-24-08026-f001:**
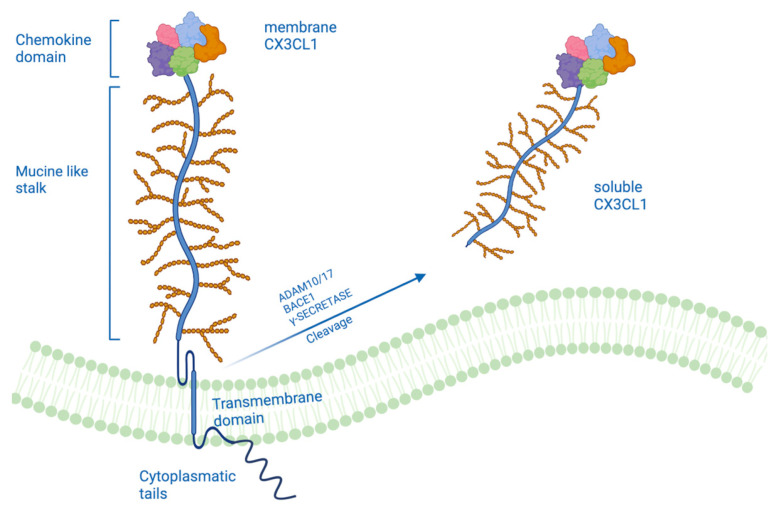
Representation of the structure of CX3CL1 composed of a chemokine domain, mucin-like stalk, a transmembrane domain and cytoplasmatic tails. The cleavage of CX3CL1 by metalloproteases and disintegrins ADAM10, ADAM 17, BACE1, and γ-secretase generates the soluble form of CX3CL1 composed only of the chemokine domain and mucin-like stalk.

**Figure 2 ijms-24-08026-f002:**
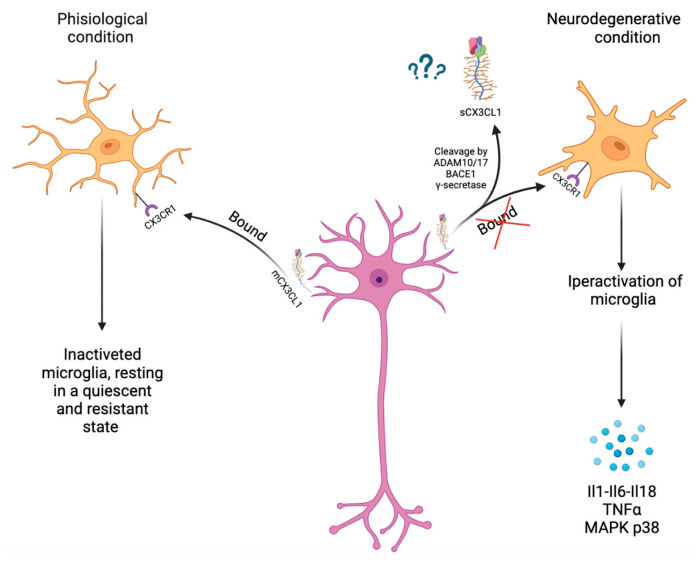
Signaling CX3CL1-CX3CR1. Physiologically mCX3CL1 bound its only receptor present on the microglia resulting in the inactivation of microglia, maintaining it in a resistant and quiescent state. In neurodegenerative disease, there is an interruption of signaling CX3CL1-CX3CR1, resulting in the hyperactivation of microglia and the production of inflammation factors. Moreover, in inflammation conditions, there is a cleavage of mCX3CL1 resulting in the formation of the soluble form of CX3CL1.

**Figure 3 ijms-24-08026-f003:**
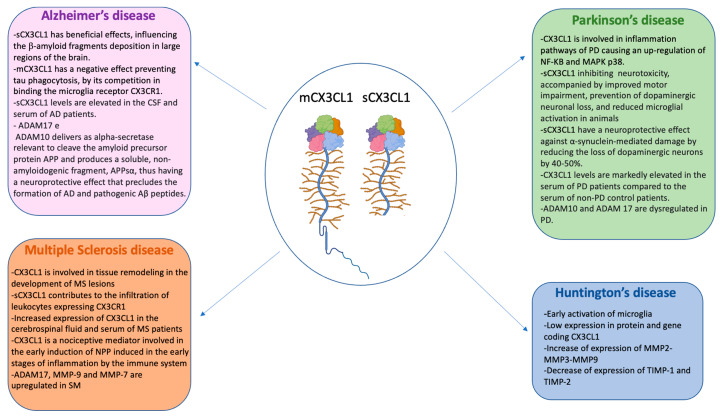
Schematic representation of the role of CX3CL1 and metalloproteases in neurodegenerative diseases.

## Data Availability

Not applicable.

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
