# Peer review of "The Cytokine CX3CL1 and ADAMs/MMPs in Concerted Cross-Talk Influencing Neurodegenerative Diseases"

_ijms, 2023, doi:10.3390/ijms24098026_

Round 1

Reviewer 1 Report

The authors in the manuscript entitled “The cytokine CX3CL1 and ADAMs/MMPs in a concerted cross talk influencing neurodegenerative diseases” have explained that the dual roles of CX3CL1 and ADAMs/MMPs in different neurodegenerative diseases, such as Alzheimer's disease (AD), Parkinson's disease (PD), Huntington's disease (MH), and multiple sclerosis (MS), are investigated.

The paper is written concisely and briefly and provide significant detail on the topic which is scientifically sound. The manuscript has been written well and the content is comprehensive.

There are some issues with this article, if these issues are going to resolve then the quality of the paper is suitable for publication.

1)             In the part of the introduction, it should be crisp and brief about the focused study with recent reference citation.

2)             Recent references should be included.

3)             There are a few typos and English and grammar errors that should be rectified.

To conclude, this is a well written and comprehensive article that makes a useful contribution to the field of neurodegenration research. The quality of the article is suitable for publication in the present form after minor revision.

Author Response

On the questions asked by the reviewer:

1)             In the part of the introduction, it should be crisp and brief about the focused study with recent reference citation. It has been done

2)             Recent references should be included.  It has been done

3)             There are a few typos and English and grammar errors that should be rectified.  The text has been further checked for typos and grammatical errors 

Reviewer 2 Report

This review is topical and of interest to the field. I have no comment or concerns.

Author Response

Thank you for the revision

Reviewer 3 Report

This well-organized review article talked about roles of CX3CL1 and several proteases in neurodegenerative diseases, providing very useful information in this important field.

However, many typos require corrections. Some examples are below.

Page 1, line 15, was “on the other hand” missing before “they target cytokines”?

Page 2, line 51, where’s the full name of ADAM?

Page 5, line 221, should “are cleaved ADAM17” be “are cleaved by ADAM17”?

Page 6, Figure 2, “Iperactivation” should be “overactivation”?

Please correct the above and other potential typos.

Author Response

On the questions asked by the reviewer:

Page 1, line 15, was “on the other hand” missing before “they target cytokines”? It has be done

Page 2, line 51, where’s the full name of ADAM? It has be done

Page 5, line 221, should “are cleaved ADAM17” be “are cleaved by ADAM17”? It has be done

Page 6, Figure 2, “Iperactivation” should be “overactivation”? It has be done

Please correct the above and other potential typos. It has be done